# Deciphering the role of the lncRNA *TRIBAL* in hepatocyte models

Sébastien Soubeyrand[1]*, Paulina Lau[1], Ruth McPherson[1,2]*

**1** Atherogenomics Laboratory, University of Ottawa Heart Institute, Ottawa, Canada, **2** Division of Cardiology, Ruddy Canadian Cardiovascular Genetics Centre, University of Ottawa Heart Institute, Ottawa, Canada

* ssoubeyrand@ottawaheart.ca (SS), rmcpherson@ottawaheart.ca (RM)

## Abstract

We recently reported that the long non-coding RNA *TRIBAL/TRIB1AL* was required to sustain key hepatocyte functions. Here, we identify HepaRG cells as a model for studying *TRIBAL* and provide additional validation and functional insights. In contrast to HepG2 and HuH-7 cells, differentiated HepaRG cells showed similarities to primary hepatocytes in response to *TRIBAL* suppression. *TRIBAL* suppression was associated with reduced HNF4A and MLXIPL abundance in hepatocytes and HepaRG cells. *TRIBAL* targeting using a panel of cognate antisense oligonucleotides confirmed specificity. A comparison of *TRIBAL*-suppressed hepatocyte and HepaRG transcriptomics identified extensive functional overlap. Biological ontologies associated with key hepatic metabolic functions were predicted to be inhibited in both models. Comparative analyses with *TRIB1*-suppressed HepaRG cells, a central metabolic regulator vicinal to *TRIBAL*, also revealed extensive functional congruence with *TRIBAL*. Interestingly, *TRIBAL* transduction failed to restore function in *TRIBAL*-suppressed cells, which may be linked to structural differences, as supported by contrasting RNAse R sensitivities between the endogenous and transduced forms. In summary, these findings support the use of HepaRG cells as an experimental model to study *TRIBAL* and underscore its importance in regulating key hepatocyte genes essential for metabolic function.

## Introduction

The 8q24.13 chromosomal region is associated with plasma lipid levels, hepatic steatosis, and risk of coronary artery disease (CAD). Initially attributed to its gene neighbor TRIB1, a major regulator of lipid homeostasis, we recently demonstrated that the genetic correlation was driven proximally by the gene encoding the long non-coding RNA (lncRNA) *TRIBAL,* also known as *TRIB1AL* [1]. LncRNAs form a functionally diverse class of transcripts defined by their length (>200 nucleotides) and lack of protein-coding potential [2]. Dismissed initially as biological noise, it is now recognized that lncRNAs are of fundamental importance [3]. Using embryonic stem cells and short

**Data availability statement:** The relevant gene expression datasets are accessible at the Gene Expression Omnibus (https://www.ncbi.nlm. nih.gov/geo/) as GSE61473, GSE248931 and GSE284599

**Funding:** This work was supported by the Canadian Institutes of Health Research (CIHR) FDN-154308 (RM). https://cihr-irsc. gc.ca/e/193.html The funder did not play play any role in the study design, data collection and analysis, decision to publish, or preparation of the manuscript.

**Competing interests:** The authors have declared that no competing interests exist.

hairpin RNA approaches, Guttman et al. showed nearly 10 years ago that out of a panel of 147 lncRNAs that were successfully suppressed, 93% had a significant impact on gene expression [4]. Since then, the list of lncRNA genes, which probably out-number protein-coding genes (estimates vary), has increased considerably, although very few are well characterized [5,6]. Their biogenesis resembles that of mRNAs. Like mRNAs, most lncRNAs are capped, polyadenylated, and are transcribed by RNA poly-merase II [7]. However, compared to mRNAs, they tend to be less efficiently spliced, evolve more rapidly, and are expressed at lower levels [7,8]. Interestingly, differential splicing may affect cellular distribution and confer unique and even antagonistic roles to lncRNAs [9,10]. LncRNAs are functionally diverse and perform numerous molecular functions in all cellular compartments, including chromatin regulation, signaling, scaf-folding, translation regulation and microRNA sponging [10–13].

We recently demonstrated that suppression of *TRIBAL* in primary hepatocytes potently reduced the expression of several major lipid regulators [1]. However, the underlying mechanisms were not explored due to technical limitations, primarily the human-specific expression of *TRIBAL* and the lack of a suitable model system. Indeed, the limited supply, high cost, and inter-donor variations of primary hepato-cytes pose significant challenges, necessitating alternative models. We previously showed that the commonly used immortalized hepatocyte-like cell models, namely HepG2 (hepatoblastoma) and HuH-7 (hepatocarcinoma), were largely unresponsive to *TRIBAL* targeted intervention, specifically *TRIBAL* overexpression (in HepG2) and *TRIBAL* suppression (in HuH-7) [14].

Here, we continue our investigation into *TRIBAL*. First, we identify HepaRG cells as a model system amenable to *TRIBAL* interrogation. HepaRG is a human cell line that can be maintained for several passages and differentiated *in vitro* into a mixture of hepatocyte- and biliary-like cells under appropriate cell culture conditions [15,16]. Although not identical to primary hepatocytes, HepaRG cells more closely resem-ble hepatocytes than HepG2s [17,18]. We now demonstrate that, similar to primary hepatocytes, *TRIBAL* expression was required to support the expression of major hepatic regulators in HepaRG cells. Specificity was ascertained with several anti-sense oligonucleotides (ASOs). *TRIBAL* suppression reduced HNF4A and MLXIPL protein abundance in primary hepatocytes and HepaRG cells. Transcriptomic data-sets from *TRIBAL*-suppressed HepaRG cells were generated with microarrays and compared with the corresponding hepatocyte data, revealing extensive similarities. Comparison of *TRIB1*- and *TRIBAL*-suppressed HepaRG cells also revealed sub-stantial functional convergence. Lastly, we report that native *TRIBAL* is considerably more resistant to RNase R digestion than the transduced transcript, suggesting that the endogenous transcript is uniquely capable of adopting complex folding.

## Methods

### Cell culture and treatments

HepaRG cells were obtained from BIOPREDIC International. Cells were seeded at 29,000 cells per cm$^2$ and maintained for 2 weeks in proliferation media, followed by 2 weeks in differentiation media. Initially, proliferation media and differentiation media were

obtained from BIOPREDIC (results shown in Fig 2). In the other experiments, homemade growth media, consisting of William's E medium supplemented with 10% fetal bovine serum, insulin (0.15 U/ml; Humalog, Eli Lilly), Hydrocortisone 21-hemisuccinate (54 μM; Cayman Chemical), and Penicillin-Streptomycin (Gibco), was used, similar to that described previously [15]. Differentiation by inclusion of 1% DMSO for 48h and of 2% DMSO for an additional 10–14 days. While side-by-side comparisons were not performed, cells grown in the BIOPREDIC media provided greater sensitivity to the *TRIBAL* antisense oligonucleotides (e.g., ASO2 in Fig 2 vs Fig 4), which could reflect differences in serum (which are tested by BIOPREDIC to confer optimal differentiation capacity) or proprietary additives leading to better terminal differentiation, a process still poorly understood. Importantly, responses were qualitatively similar (i.e.,general suppression of the TOI). Media was changed every 48–72 hours. HuH-7 (Japanese Collection of Research Bioresources Cell Bank) and HepG2 (ATCC) were maintained in DMEM supplemented with 5mM glucose, 10% fetal bovine serum, and Antibiotic–Antimycotic (Gibco). Cryopreserved primary hepatocytes (HMCPMS), media, and media supplements were obtained from ThermoFisher Scientific. Lots HU8413 (lot 2) and HU8412 (lot 1) were thawed in thawing media and seeded in 12-well plate wells (14–16 wells per vial). The media was changed to a culture medium, which was replaced every 24 hours until the transfection, where incubation was continuous in the presence of the ASOs. Experiments were initiated 4–6 days post-plating, resulting in 3 distinct replicates for each donor, as described previously [1].

### Viral particle generation

Viral particles were generated in 293FT cells with the empty PLVX plasmid, PLVX*TRIBAL1*, psPAX2, and pMD2G. Viral particles harvested during the 16–72h window post-transfection were concentrated with Lenti-X (Clontech) and stored at −80°C. Titers were determined on HEK293T cells using puromycin selection (3 μg/ml). Cells were inoculated with three multiplicities of infection (MOI).

### Western blotting

Cells were extracted for 5min on ice in lysis buffer (50mM Tris-HCl, 150mM NaCl, 1% NP40, pH 7.4) supplemented with protease and phosphatase inhibitors (PhoStop and Complete, Roche). Samples were centrifuged (2min @ 16kg), and the supernatant (20 μg of protein) was denatured in 1X Laemmli SDS-PAGE buffer (95 °C, 5min). Samples were then subjected to SDS-PAGE (10% gels) and transferred using liquid transfer (1h, 90 V) to nitrocellulose membranes. Even loading and transfer were ascertained by Ponceau staining. Blots were then destained and blocked in Intercept buffer (LI-COR) for one hour. Detection was performed using antibodies diluted in PBS/0.1% Tween. Primary antibodies (1:1000 dilution) were incubated for 16 hours, and secondary antibodies (LI-COR; 1:20,000 dilution) for 1 hour. Four 30-second washes in PBS were performed after each antibody incubation. Blots were imaged on an Odyssey Imager (LI-COR). All images were within the instrument's dynamic range and were only adjusted in contrast and intensity. Complete blots are available as Supporting information (S1 File **Uncropped blots)**.

### CRISPR and dCAS9 targeting of *TRIBAL*

Pairs of oligonucleotides matching two regions flanking exon one were inserted downstream of the U6 promoter of a trcrRNA expression plasmid (pCRU6) [19]. Cells were transfected with pCRU6 constructs and eSpCas9 a gift from Feng Zhang (Addgene plasmid # 71814) [20]. Transfections were performed in 24-well plate wells using 0.5 μg of DNA with an 8:1:1 (eSpCas9:pCRU6g1:pCRU6g2) DNA mass ratio. HepG2 and HuH-7 (40−60% confluent) cells were transfected using Extreme Gene HP (3:1, HP: DNA) or Lipofectamine 3000 (3:2:1, Lipofectamine 3000: P3000: DNA). dCAS9 activation was performed using a dual expression plasmid (pAC154-dual-dCas9VP160-sgExpression) encoding inactive CAS9 coupled to VP160 and a U6-driven sgRNA cassette, a gift from Rudolf Jaenisch obtained via Addgene (plasmid # 48240) [21]. Oligonucleotides targeting two regions within the *TRIBAL* promoter were cloned in the U6 cassette. HepG2 and HuH-7 cells were transfected for 72h with the dCas9VP160 containing sgRNA4, sgRNA5, or trcrRNA control. Oligonucleotide sequences are listed in Supporting information (S2 File Supplemental Materials**).**

## Antisense oligonucleotides

Antisense oligonucleotides (ASOs) were designed as gapmers. ASO1 (targeting exon 1) and ASO2 (targeting *TRIBAL* intron 2) were previously used to suppress *TRIBAL* expression in hepatocytes [1]. Both ASOs had directionally consistent impacts, although the effects of ASO2 tended to be more pronounced. Other ASOs were designed to mitigate off-target concerns: ASO5 and ASO6 targeting intron 2 and ASO9 and 10 targeting intron 1. All intron-targeting ASOs displayed comparable efficiencies in HepaRGs, reducing *TRIBAL* by approximately 50%, except for ASO5 which was slightly less effective. Sequences are listed in Supplemental Materials (S2 File Supplemental Materials). HepG2 and HuH-7 cells were transfected at 30%−50% confluence in 0.5 mL of media in 24-well plates with 10 nM ASO (final concentration) and 1 µL of RNAiMax. The siRNA transfection mix was prepared in 100 µL of Opti-MEM (Gibco). For HepaRG (24 well-plates, 0.5 ml per well) and primary hepatocytes (12 well-plates, 1 ml per well), transfection was performed using 60 nM ASO (final) and 2.4 µl (HepaRG) or 4.8 µl (hepatocytes) of RNAiMax in 100 µl (HepaRG) or 200 µl (hepatocytes) Optimem. Treatment was continuous for 72 hours.

## Real-Time RNA quantification PCR (RT-qPCR) and PCR

RNA was extracted from culture plates using TriPure Reagent (Roche) and isolated using Direct-Zol RNA mini prep kits (Zymo Research). RNA (0.25 µg in 10 µl) was reverse-transcribed using the Transcriptor First Strand cDNA kit (Roche), employing a 1:1 mixture of oligo dT and random hexamer primers for 1 hour. The resulting cDNA was diluted sixfold in H2O and quantified using a Light Cycler 480 with SYBR Green (Roche) and 0.5 µM primers in 384-well plates. Target of interest values were expressed relative to the corresponding peptidyl-prolyl isomerase A (PPIA) values using the ΔΔCt method. PPIA is routinely used in our laboratory as a robust housekeeping gene in hepatocyte models based on its insensitivity to *TRIBAL* (and TRIB1) ASO treatments in qRT-PCR and transcription array expression datasets (e.g., GSE284599, GSE248931, GSE61473). For PCR, cDNA (HepaRG *TRIBAL1* characterization) or whole cell lysates (genotyping of HepG2 and HuH-7 cells) was amplified using Terra PCR Direct (Takara Bio). A 65−54 touchdown (11 cycles) was followed by isothermic (54 °C) amplification rounds (29 cycles), using 1 min extension times. Oligonucleotides are detailed in Supporting information (S2 File Supplemental Materials).

## RNAse R digestion

Total RNA was purified using Tri-Reagent (Roche) and Direct-Zol RNA miniprep kits (Zymo Research). RNA (500 ng) was then incubated with 0.5 µl of RNase R (Ambion) for 2 hours at 37°C in a total volume of 25 µl. Samples were then reverse-transcribed as described above and analyzed by qPCR. Relative sensitivity was calculated using the ΔΔCt method by comparing the abundance of mock-incubated RNA to the matching RNase R-treated value.

## Transcriptome clustering analyses

Three transcription array datasets consisting of ASO-treated Hepatocytes and HepaRG cells were used: GSE61473 (Hepatocytes, *TRIB1* ASO1, 48 h), GSE248931 (Hepatocytes, *TRIBAL* ASO1 and ASO2, 72 h), and GSE284599 (HepaRG, *TRIBAL* ASO2 and *TRIB1* ASO2, 72 h). Data are available at the Gene Expression Omnibus depository. The entire list of mappable Entrez Gene IDs was used for GSEA, using the Gene Ontologies (Biological noRedundant), through WebGestalt. Rather than relying on an arbitrary significance threshold (as in overrepresentation analyses), GSEA employs a ranking approach that aggregates incremental changes in transcripts within categories to better capture enrichment patterns within a dataset. GSEA through WebGestalt calculates an enrichemment score and generates p-values using 1000 permutations. Output was limited to a maximum of 200 terms, and highly significant (p < 0.01) terms were retained. Comparative analyses were conservatively performed on FDR-significant terms to minimize false positives. Thus, the actual

overlap is probably underestimated. The clustering of enriched GO terms based on semantic similarity was performed using Revigo, using the default parameters [22]. Visualization was performed via Cytoscape [23].

## Results

### The TRIBAL locus, but not its transcript, is required for growth of HepG2 and HuH-7 cells

*TRIBAL* suppression in primary hepatocytes resulted in pervasive transcriptome-wide changes in primary hepatocytes [1]. Earlier efforts using an antisense oligonucleotide (ASO1) to suppress *TRIBAL* in HuH-7 and *TRIBAL* overexpression in HepG2 cells failed to uncover consistent transcriptome-wide changes [14]. Examination of a subset of transcripts of interest (TOIs), selected based on their response to *TRIBAL* suppression and functional importance in our previous work in hepatocytes, revealed little or no consistent effects in response to suppression using two distinct *TRIBAL*-targeting ASOs (Fig 1) [1]. Similarly, increased *TRIBAL* expression via activating CRISPR (Clustered Regularly Interspaced Short Palindromic Repeats)-Cas9 did not increase the TOI expression (S1 Fig). Indeed, some inhibitory trends were visible in HuH-7 (S1 Fig). Thus, these model systems were unsuitable for informing the physiological role of TRIBAL. Interestingly, the locus still provided some benefit to these cells: HepG2 and HuH-7 cell pools harboring a CRISPR-Cas9-mediated *TRIBAL* locus deletion were gradually depleted (S2 Fig).

### The HepaRG model partially recapitulates the dependence of primary hepatocytes on *TRIBAL*

Looking for a suitable cell model alternative, HepaRG cells were next examined. HepaRG cells acquire hepatocyte-like gene expression and morphology upon differentiation and resemble hepatocytes more than HepG2 cells [17,24,25]. Unlike HepG2 and HuH-7 cells, *TRIBAL* suppression in differentiated HepaRG cells reduced the expression of targets of interest previously shown to be impacted in primary hepatocytes (**Fig 2**) [1]. As we reported in primary hepatocytes,

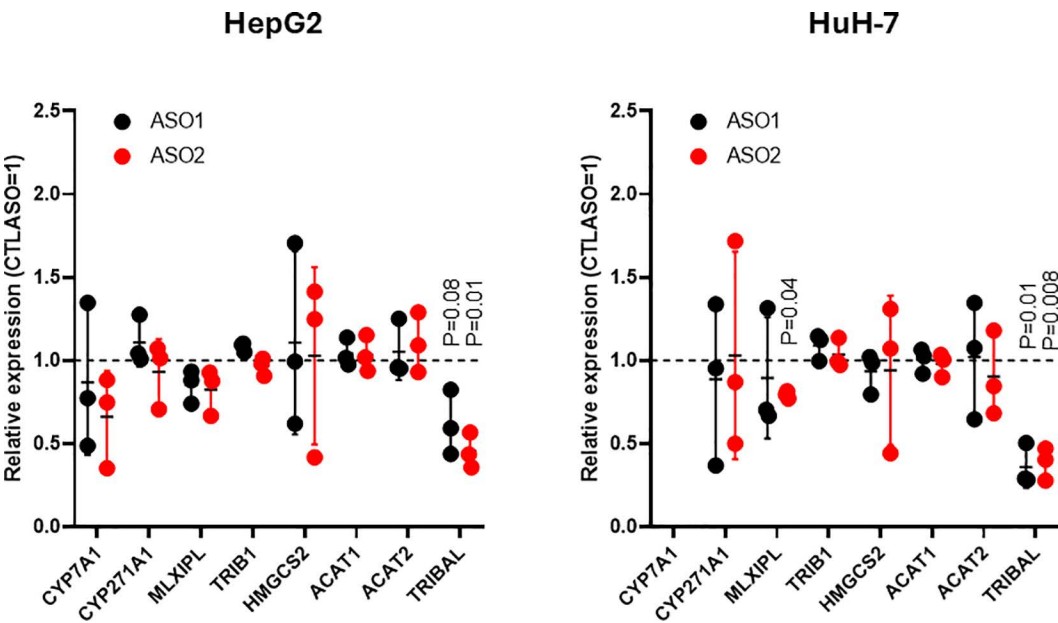

**Fig 1. *TRIBAL* suppression in HepG2 and HuH-7 models.** HepG2 and HuH-7 cells were treated with antisense oligonucleotides (ASO1, ASO2, or a non-target control (CTLASO)) for 72 h. Transcript abundance was measured by qRT-PCR. The values shown are from 3 biological replicates. Error bars are the mean ± S.D. CYP7A1 levels in HuH-7 cells were undetectable in this set of experiments. Statistical significance was assessed using a one-sample t-test with a theoretical control value of 1 (CTLASO = 1) in Prism. P values approaching (p ≤ 0.1) or surpassing nominal significance are shown.

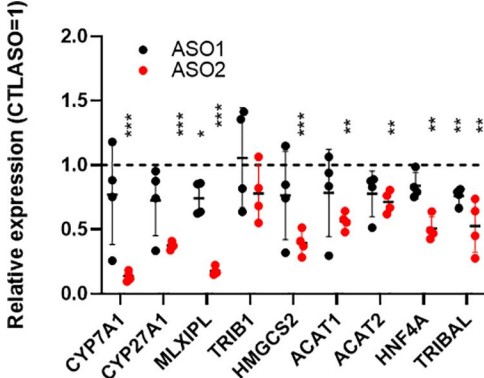

**Fig 2. Suppression of *TRIBAL* in HepaRG on metabolic transcript panel.** qRT-PCR analysis of HepaRG cells treated for 72 h with 2 TRIBAL ASOs. Two independent suppression rounds (for a total of 4 biological replicates) were performed. Each point represents a different biological replicate, obtained by transfections staggered by 24 hours (days 12-15 post-differentiation start). Transcript abundance is expressed relative to the CTLASO value as determined by qRT-PCR. Averages ± S.D. are shown. Statistical significance was tested using a one-sample t-test using a theoretical control value of 1 (1 = CTLASO value). *, $p < 0.05$; **, $p < 0.01$; ***, $p < 0.001$.

suppression was particularly evident with ASO2, although a concordant but significantly weaker pattern was observed with ASO1 [1]. Moreover, *hepatocyte nuclear factor 4 alpha (HNF4A)* expression was also reduced, consistent with our previous finding that HNF4 function is impaired in *TRIBAL*-suppressed hepatocytes. Immunoblotting confirmed the reduction in HNF4A and MLX Interacting Protein Like (MLXIPL) protein abundance in both primary hepatocytes and HepaRG, although the decrease in HepaRG was observed only with ASO2 (**Fig 3**).

### Identification of additional *TRIBAL*-suppressing antisense oligonucleotides

Next, we examined whether the stronger suppressive ability of ASO2 (compared to ASO1), also observed in hepatocytes, resulted from possible off-targeting [1]. To alleviate off-targeting concerns, we searched for additional *TRIBAL*-targeting

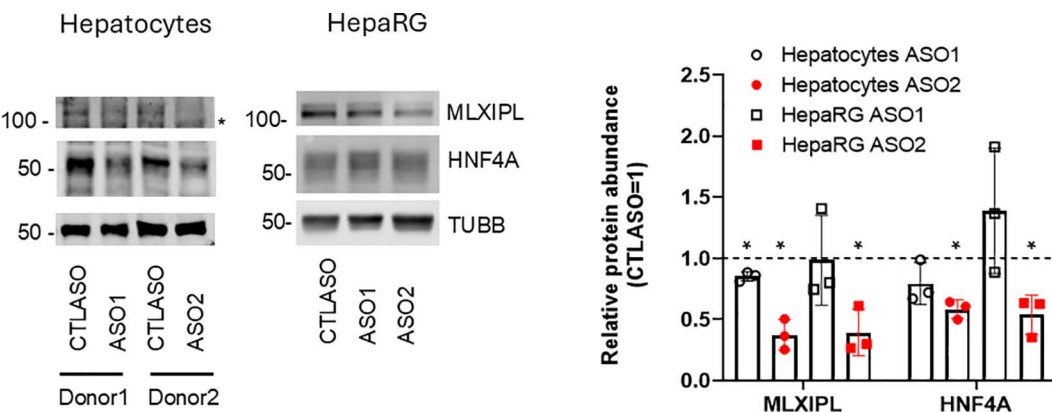

**Fig 3. Impact of *TRIBAL* suppression on HNF4A and MLXIPL protein abundance in hepatocytes and HepaRG cells.** Hepatocytes and HepaRG cells were suppressed for 72 h with either ASO1 or ASO2, as indicated. Whole-cell lysates were analyzed by western blot using the indicated antibodies. The migration of the protein marker is shown on the left. Quantification of 3 biological repeats (normalized to Tubulin beta (TUBB)) and expressed relative to the matching CTLASO value, is shown on the right, (± S.D.). Statistical significance was tested using a one-sample t-test with a theoretical control value of 1 (1 = CTLASO value). *, $p < 0.05$. Detection was performed sequentially, first with HNF4A and then MLXIPL/TUBB for hepatocytes, and MLXIPL and then HNF4A/TUBB for HepaRG cells. * indicates a non-specific band resulting from sequential probing.

ASOs. After an exploratory screen of 5 additional ASOs in HepaRG, 4 showed evidence of suppression (2 vs. intron 1 and 2 vs. intron 2) (Fig 4). All but one ASO resulted in reduced *TRIBAL* abundance (although statistical significance was observed with only two ASOs in this suppression round) and, importantly, led to comparable suppression of several of the targets of interest. These findings confirmed that these ASOs specifically target *TRIBAL*.

## TRIB1 suppression partially phenocopies *TRIBAL* suppression in HepaRG and hepatocytes

TRIB1 is another important regulator of lipid metabolism. Its proximity to *TRIBAL* suggests that *TRIBAL* and TRIB1 may be functionally intertwined, as observed for several lncRNAs and their proximal protein-coding gene [26]. Indeed, we previously demonstrated that *TRIB1* suppression increased *TRIBAL* expression in primary hepatocytes [14]. Moreover, *TRIBAL* suppression was generally associated with lower *TRIB1* expression, although statistical significance was achieved only for ASO2 in one series of experiments (Fig 2, 4). Thus, *TRIBAL* may promote *TRIB1* expression, in turn facilitating TRIB1 function. First, the functional overlap between *TRIBAL* and TRIB1 was explored using publicly available data of ASO-treated hepatocytes. Targeting *TRIB1* or *TRIBAL* in hepatocytes resulted in a significant reduction of the TOIs, indicative of functional convergence (Fig 5A). Consistency with HepaRG was then tested by qRT-PCR, using *TRIBAL* ASO2 and two distinct *TRIB1*-targeting ASOs in HepaRG cells. The findings were overall consistent with those in primary hepatocytes, with *TRIB1* suppression resulting in similar (*TRIB1* ASO2) or greater (*TRIB1* ASO1) suppression than *TRIBAL* ASO2 in our transcript panel (Fig 5B). A notable exception was *CYP7A1*, which was responsive to TRIB1 suppression only in HepaRG cells.

## Undifferentiated but confluent HepaRG cells are unaffected by *TRIBAL* targeting

Although differentiation of HepaRG post-confluence is essential for the complete acquisition and maximal expression of hepatocyte-like traits, undifferentiated HepaRG cells express several hepatocyte-specific genes [15]. Consistent with a possible function, *TRIBAL* was expressed at levels comparable to/ differentiated cells, although considerable inter-experimental variation, not unique to HepaRG cells, was evident (S3 Fig). To examine whether the *TRIBAL* function required complete differentiation, suppression was repeated in confluent but undifferentiated cells (10–14 days post-seeding). Unlike differentiated

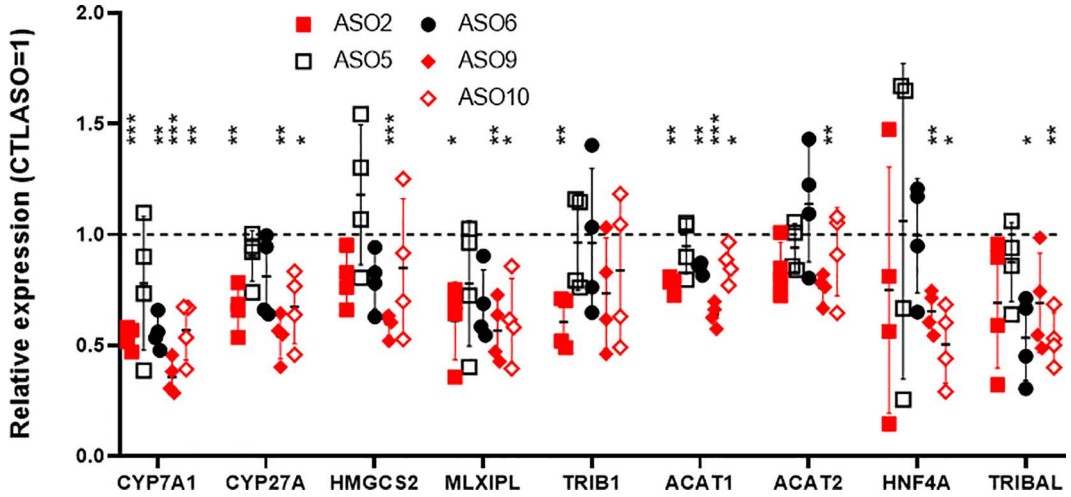

**Fig 4. Screening of additional ASOs on HepARG. qRT-PCR analysis of HepaRG cells treated for 72 h with a panel of antisense oligonucleotides.** Each point represents a different biological replicate, obtained by transfections staggered by 24 h (days 11-14 post-differentiation start). Symbols in red and black are ASO targeting intron 1 and 2, respectively. Transcript abundance is expressed relative to the CTLASO value. Averages ± S.D are shown. Statistical significance was tested using a one-sample t-test with a theoretical control value of 1 (1 = CTLASO value). *, p < 0.05; **, p < 0.01; ***, p < 0.001.

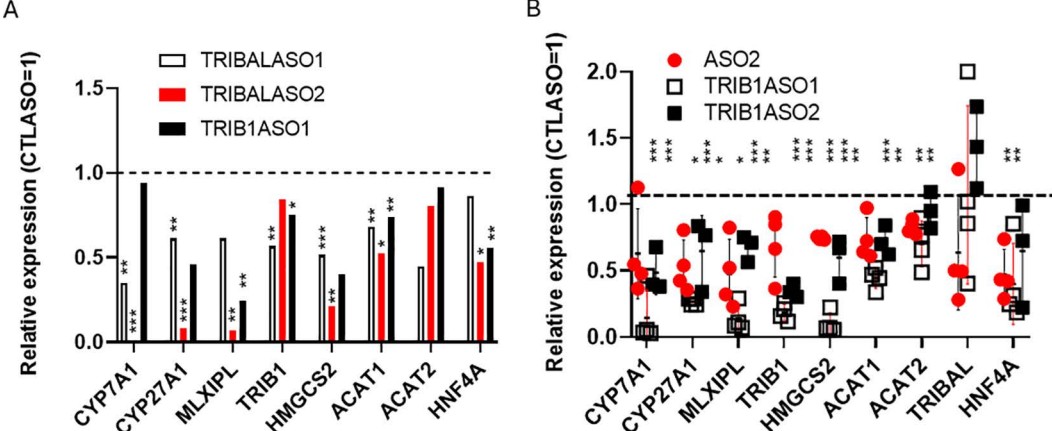

**Fig 5. *TRIB1* and *TRIBAL* are necessary for sustaining the expression of TOIs in primary hepatocytes and HepaRG cells.** A, the impact of ASOs in primary hepatocytes. Data from the Transcription Analysis Console corresponding to GSE248931 and GSE61473. Note that *TRIBAL* is not represented on these arrays. Statistical analysis from the Transcriptome Analysis Console (ebayes ANOVA). B, impact of ASOs in HepaRG cells. Values from qRT-PCR analysis of HepaRG cells treated for 72 h with *TRIBAL* (ASO2) or *TRIB1* (*TRIB1* ASO1, *TRIBAL* ASO2). Each point represents a different biological replicate, obtained by transfections staggered by 24 hours (days 11-14 post-differentiation start). Transcript abundance is expressed relative to the CTLASO value. Averages ± **S**.D. are shown. Statistical significance was tested using a one-sample t-test using a theoretical control value of 1 (1 = CTLASO value). *, $p < 0.05$; **, $p < 0.01$; ***, $p < 0.001$.

HepaRG, *TRIBAL* targeting consistently failed to reduce *TRIBAL* abundance (Fig 6A). Indeed, *TRIBAL* expression was occasionally increased. By contrast, *TRIB1* silencing resulted in cognate suppression, confirming efficient cellular targeting (Fig 6B). Thus, *TRIBAL* does not appear to be functional in undifferentiated HepaRG cells or may not be effectively targeted. We reasoned that the considerable variation in *TRIBAL* and TOI abundance might reflect a potential causal relationship. Indeed, when the TOI values were plotted relative to *TRIBAL* abundance (normalized to the NT values), a clear pattern emerged: while most of the TOIs were not correlated to *TRIBAL* abundance, arguing for a dysfunctional *TRIBAL* axis, *TRIB1* levels uniquely clustered around 1, suggesting changes in *TRIBAL* and *TRIB1* were correlated (Fig 6C). Regressing *TRIB1* against *TRIBAL* relative expression confirmed a monotonic relationship (Fig 6D). This tight correlation is consistent with the presence of mechanisms driving the coordinated expression of *TRIB1* and *TRIBAL*, at least in undifferentiated HepaRG cells.

### Comparison of *TRIBAL*-suppressed hepatocytes and HepaRG transcriptomes by Gene Set Enrichment Analysis

A transcriptomic investigation was then conducted to clarify the role of *TRIBAL* in primary hepatocytes and HepaRG cells and to delineate the functional footprints of TRIB1 and *TRIBAL*. To this end, *TRIBAL*- and *TRIB1*-ASO2-treated HepaRG samples were selected for transcriptome-wide expression analysis by microarrays. First, a transcript-level comparison was made between *TRIBAL*-suppressed HepaRG cells and hepatocytes. Compared to hepatocytes, *TRIBAL* suppression in HepaRG had a reduced impact: transcripts were generally less affected (curbed fold-changes) and ~5 times fewer transcripts reached nominanl significance (S4 Fig). Importantly, approximately half of the nominal HepaRG transcripts were also significant in hepatocytes, indicating considerable convergence (p = 0.001, Jaccard similarity test) (S5 Fig).

To translate these differences into a biological understanding, the transcriptomes were then clustered to biological ontologies using Gene Set Enrichment Analysis (GSEA). The previously reported hepatocyte analysis was repeated to account for updates to the pipeline and dataset. The comparison of FDR significant terms revealed extensive overlap, predominantly in metabolic processes (S5 Fig, S1 and S2 Tables). Notably, the effects on the most impacted pathways were predominantly suppressive (negative effect size) in both models, indicating that these processes are curtailed in *TRIBAL*-targeted cells (Table 1). Examining terms specific to either cell type revealed growth and proliferation-related

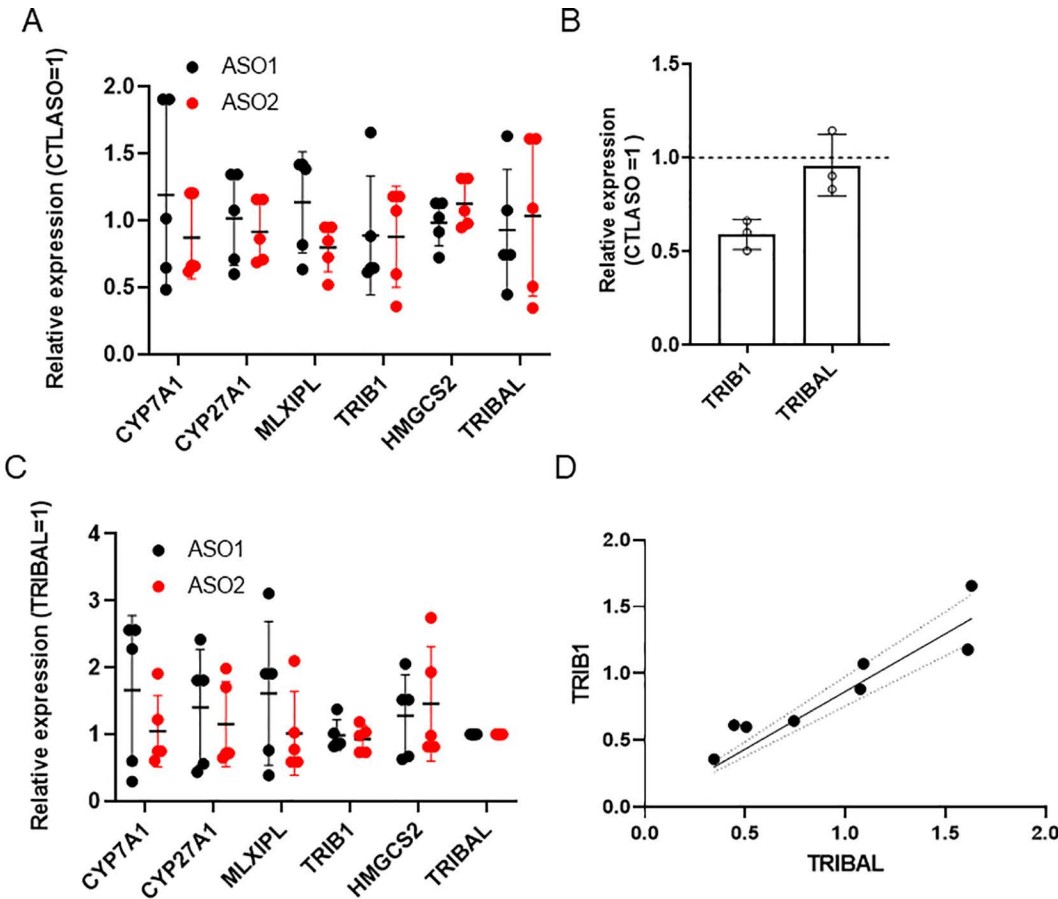

**Fig 6. Response of confluent but undifferentiated HepaRGs to *TRIBAL* suppression.** A, undifferentiated but confluent HepaRG cells were treated with *TRIBAL* ASO1, ASO2, or CTLASO for 72 h. RNA was then isolated and quantified using qRT-PCR. In B, cells were treated with *TRIB1* ASO2 for 72 h. In C and D, data from A was expressed relative to the *TRIBAL* level. In D, *TRIB1* values from C are expressed relative to *TRIBAL*.

**Table 1. Comparison of shared and highly FDR significant ontologies in Hepatocytes and HepaRG treated with *TRIBAL* ASO2.**

| | Hepatocytes | | HepaRG | |
|---|---|---|---|---|
| | NES | FDR | NES | FDR |
| alcohol metabolic process | −3.67 | 0 | −2.592 | 2.7E-03 |
| fatty acid metabolic process | −3.64 | 0 | −2.988 | 0 |
| olefinic compound metabolic process | −3.59 | 0 | −2.506 | 5.2E-03 |
| steroid metabolic process | −3.26 | 1.1E-04 | −2.388 | 6.4E-03 |
| secondary metabolic process | −3.14 | 9.4E-04 | −2.354 | 8.2E-03 |
| sulfur compound metabolic process | −3.09 | 1.4E-03 | −2.827 | 0 |
| cellular ketone metabolic process | −3.09 | 1.3E-03 | −2.355 | 9.0E-03 |
| nucleoside bisphosphate metabolic process | −2.91 | 7.8E-03 | −2.643 | 2.6E-03 |

Table showing shared and highly significant (q<0.01) FDR ontologies identified by GSEA in HepaRG and hepatocytes. Organized by incremental effect size in hepatocytes. NES, normalized effect size. For a complete list of impacted ontologies, see S1–S2 Tables

ontologies in HepaRG cells, whereas more metabolic ontologies were observed in primary hepatocytes (S3 Table). Within shared ontologies, term clustering revealed a network of six terms centered on phospholipid metabolism (S5 Fig).

**Transcriptome-wide impact of *TRIB1* and *TRIBAL* suppressionsin HepaRG**

The functional interplay between *TRIBAL* and *TRIB1* was then examined by comparing their suppression in HepaRG cells. At the transcript level, nearly half of the *TRIBAL* ASO2-impacted transcripts were also nominally affected in *TRIB1* ASO2-treated cells (S6 Fig). GSEA of *TRIB1*-targeted HepaRG cells identified 47 FDR-significant ontologies, of which 18 were similarly impacted in *TRIBAL*-suppressed cells (S6 Fig, Table 2, S4 Table). This shared group included several core metabolism terms, some showing a robust statistical association, including lipid and steroid-related ontologies (Table 2, S5 Table).

To complement these findings, GSEA was repeated using the *TRIB1* (rather than the corresponding CTLASO) expression values as the *TRIBAL* background, which the concurrent suppression design permitted. By aggregating transcript-level differences between *TRIB1*- and *TRIBAL*-targeted populations, we reasoned that this approach should enable the estimation of the relative contribution of *TRIB1* and *TRIBAL* to any given ontology. Clustering identified 11 FDR-significant enriched categories, including a single metabolic ontology, "glucose-6-phosphate metabolic process", which was negatively enriched relative to *TRIB1*, suggesting that these functions are more inhibited in *TRIBAL*-suppressed cells (S6 Table). In contrast, *TRIBAL* suppression was associated with the positive enrichment of a few FDR-significant terms linked to inflammation ("response to type II interferon," "humoral immune response," and "response to protozoan"), consistent with the establishment of a more inflammatory environment. Notably, most of the major ontologies identified in the individual GSEA analyses were not significantly different in this analysis, indicating a generally concordant impact.

**Table 2. Biological processes enriched in *TRIBAL*- and *TRIB1*-suppressed HepaRG cells.**

| | HepaRG *TRIBAL* | | HepaRG TRIB1 | |
|---|---|---|---|---|
| | NES | FDR | NES | FDR |
| fatty acid metabolic process | −3.0 | 0 | −3.4 | 0 |
| sulfur compound metabolic process | −2.8 | 0 | −2.3 | 1.2E-02 |
| nucleoside bisphosphate metabolic process | −2.6 | 2.6E-03 | −2.9 | 0 |
| alcohol metabolic process | −2.6 | 2.7E-03 | −2.8 | 0 |
| olefinic compound metabolic process | −2.5 | 5.2E-03 | −2.3 | 1.3E-02 |
| steroid metabolic process | −2.4 | 6.5E-03 | −2.8 | 0 |
| cellular ketone metabolic process | −2.4 | 9.0E-03 | −2.3 | 1.2E-02 |
| fatty acid derivative metabolic process | −2.3 | 1.1E-02 | −2.6 | 1.7E-03 |
| small molecule catabolic process | −2.3 | 1.2E-02 | −2.2 | 1.9E-02 |
| hormone metabolic process | −2.2 | 1.3E-02 | −2.0 | 3.2E-02 |
| isoprenoid metabolic process | −2.2 | 1.6E-02 | −2.2 | 2.1E-02 |
| organic acid biosynthetic process | −2.2 | 1.7E-02 | −2.4 | 1.2E-02 |
| platelet-derived growth factor receptor signaling pathway | −2.2 | 1.8E-02 | −2.0 | 3.2E-02 |
| post-embryonic development | −2.1 | 2.3E-02 | −2.2 | 1.9E-02 |
| lipoprotein metabolic process | −2.0 | 3.6E-02 | −2.2 | 1.8E-02 |
| smoothened signaling pathway | −2.0 | 3.6E-02 | −1.9 | 4.4E-02 |
| lipid modification | −2.0 | 3.4E-02 | −2.2 | 1.8E-02 |
| organic hydroxy compound catabolic process | −2.0 | 3.8E-02 | −1.9 | 4.9E-02 |

Impacted transcripts were mapped by GSEA to biological ontologies (non-redundant) via WebGestalt. FDR significant biological ontologies common to *TRIBAL*- and *TRIB1*-suppressed cells are shown below, organized by increasing effect size (HepaRG values). For a complete list of impacted ontologies in *TRIBAL* and *TRIB1*-suppressed cells, see S2 and S4 Tables.

## Transduced *TRIBAL1* cannot rescue *TRIBAL* suppression

We previously demonstrated that *TRIBAL1* overexpression had no impact on primary hepatocytes, implying that transduced *TRIBAL1* might be dysfunctional or insufficient [1]. Importantly, *TRIBAL1* was the only splice variant of *TRIBAL* identified in primary hepatocytes by Rapid amplification of cDNA ends (RACE) and consistent results were obtained by PCR and restriction digest in HepaRG cells (S7 Fig). As with primary hepatocytes, expression of the gene panel was not affected in HepaRG transduced with *TRIBAL1* (S8 Fig). To test whether *TRIBAL1* is functional, a HepaRG pool stably expressing *TRIBAL1* was targeted using the intron-specific ASO2, thus not targeting transduced *TRIBAL1*. We reasoned that the presence of *TRIBAL1* should offer protection from ASO2 only if *TRIBAL1* is functional. Indeed, whereas the endogenous *TRIBAL* in the transduced PLVX controls was highly sensitive to ASO2, resulting in greater than 90% suppression of *TRIBAL*, overexpressed *TRIBAL*1 was unaffected (S9 Fig). Importantly, *TRIBAL1* did not protect against ASO2. As off-target effects of ASO2 were deemed improbable, given the consistent impact of several other cognate ASOs (Fig 4), these experiments suggested that overexpressed *TRIBAL1* was dysfunctional.

## Endogenous, but not transduced *TRIBAL*, is resistant to RNAse R

We hypothesized that *TRIBAL1* dysfunction may stem from improper folding or maturation. The native expression locus may impart unique features that the lentiviral insertion sites may not provide. Structural differences were probed using RNAse R, a structure-sensitive RNA exonuclease. Strikingly, recombinant *TRIBAL1* was highly sensitive to RNAse R, on par with the PPIA mRNA. By contrast, native *TRIBAL*, like U1, a highly structured snRNA, showed a near-complete resistance to RNAse R (**Fig 7**). Thus, the native locus may be key in conferring functionality to *TRIBAL*, possibly by imparting unique folding attributes.

## Discussion

TRIB1 and *TRIBAL* are both required to support hepatocyte function. As demonstrated for *TRIBAL*, we previously demonstrated that *TRIB1* suppression reduced *HNF4A* expression and function in primary hepatocytes [27]. Given the central role of HNF4A in establishing and maintaining liver function, substantial functional convergence between *TRIB1* and *TRIBAL* is to be expected. Moreover, the subset of transcripts populating our TOI list, defined in our previous work based

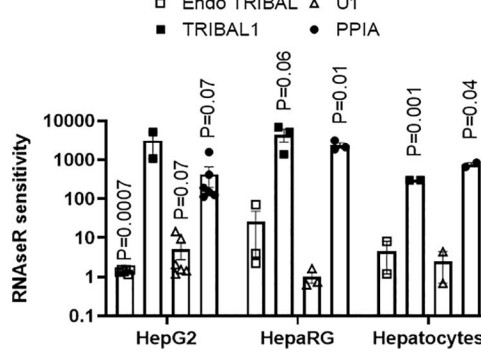

**Fig 7. The native locus is resistant to RNAse R.** RNA from HepaRG, Hepatocytes, or HepG2 cells was isolated and subjected to mock or RNAse R digestion. pLVX or pLVX*TRIBAL1* transduced cells were used for Endo or *TRIBAL1* experiments, respectively. The *U1* and *PPIA* values are from the pLVX transduced cells. RNAse R sensitivity is the ratio of the mock signal divided by the RNAse R values. Statistical significance was tested using a one-sample t-test and a theoretical control value of 1 (i.e., complete resistance to RNAse **R**). A one-tailed analysis was performed under the assumption that RNAse R treatment could only reduce RNA abundance. Error bars represent the **S**.D. Only p ≤ 0.1 are shown. Each point represents a biological repeat.

on the effects of *TRIBAL* suppression, was also inhibited by *TRIB1* suppression. Transcriptomic enrichment analyses are consistent with this convergence, although unique contributions were identified. However, one must interpret these "unique" contributions cautiously, as the residual expression may suffice to ensure partial functionality. Thus, clustering approaches likely underestimate the actual participation of *TRIBAL* and TRIB1.

The evidence of functional convergence should not be misconstrued as evidence of obligate functional epistasis. For instance, *TRIB1* suppression was associated with reduced expression of the TOIs without *TRIBAL* reduction in HepaRG cells (e.g., Fig 5B). Similarly, *TRIBAL* suppression occasionally led to reduced TOI expression with minimal *TRIB1* impact (e.g., **Fig 2**). These results suggest that TRIB1 and *TRIBAL* can independently impact the TOIs, although further experiments will be necessary to corroborate and substantiate these findings. This functional independence contrasts with a previous model, whereby *TRIBAL* was proposed to regulate liver function through TRIB1, whose role in hepatic function was already well established. That hypothesis was based on early evidence indicating that lncRNAs operate locally and act as enhancers [28,29]. However, this model needs to be reassessed in light of recent transcriptome-wide interventions examining the contribution of proximal protein-coding and non-coding genes. Based on essentiality criteria, 778 lncRNAs required for proliferation in at least 1 of 5 cell lines were identified [30]. Remarkably, these were prone to operate independently of neighboring protein-coding genes, highlighting their functional independence. How *TRIBAL* and TRIB1 converge to support the expression of these important regulators remains an important question that future work will aim to elucidate.

Here we used differentiated 2D HepaRG cultures as models for *TRIBAL* studies. Using an algorithm prioritizing a panel of liver-specific transcripts, Kim et al. have shown that 3D and 2D cultures of HepaRG cells exhibit 59% and 41% liver similarity, demonstrating that both models are good but imperfect approximations of hepatocytes. However, both were superior to hepatocyte-like cells derived from iPSC [24]. In this work, significant differences in TOI expression were observed across the hepatoma models and between hepatoma models and primary hepatocytes, underscoring that no single model perfectly mimics primary hepatocytes (S10 Fig). Thus, the sensitivity of HepaRG cells to the ASOs cannot be explained solely by transcript abundance.

Interestingly, HepaRG cells grown in 2D can differentiate (and transdifferentiate) into hepatocyte- and biliary-like cells in approximately equal proportions [31]. It will be interesting to examine the implications of *TRIBAL* in this process, as lncRNAs can regulate pluripotency and differentiation [4]. The role of *TRIBAL* in biliary cells remains unexplored, and its absence may contribute to the weaker impact of *TRIBAL* ASOs on HepaRG cells compared to primary hepatocytes, as identified by our enrichment analyses. Alternatively (or in addition to), the more modest effect of *TRIBAL* suppression in HepaRG cells could be due to the incomplete acquisition of hepatocyte traits.

Compared to ASO2, ASO1 treatment produced a weaker transcriptional response in HepaRG, thus recapitulating our observation in hepatocytes. Moreover, we observed a similar pattern at the protein level, where MLXIPL and HNF4A were more impacted by ASO2 in both hepatocytes and HepaRG cells. However, our previous work uncovered considerable overlap between the transcriptome-wide impacts of the 2 ASOs. The reasons behind this difference in potency are still unclear. The reduced effects of ASO1 may be related to its slightly lower *TRIBAL* knock-down efficacy (see Fig 2). However, we previously showed that targeting hepatocytes with a distinct ASO against exon 1 suppressed the TOIs to the same extent as ASO1, despite achieving a greater *TRIBAL* reduction [1]. An alternative explanation may be that the more effective intron-targeting ASOs affect *TRIBAL* variants devoid of exon 1. However, we have not detected exon 1-free variants by 5' or 3' RACE in HepG2 cells or hepatocytes [1,14]. Rather, as previously reported, the greater impact might be due to the targeting of emerging RNA transcripts leading to premature transcription termination [32]. In this model, *TRIBAL* transcription *per se* contributes significantly to its biological role. In addition to the folding hypothesis discussed further below, this model may account for the inability of transduced *TRIBAL* to protect against *TRIBAL* suppression.

It is unclear how and why undifferentiated HepaRG cells resisted *TRIBAL* suppression. Control *TRIB1* suppression was effective and reproducible, arguing that the ASOs were delivered appropriately and that the RNA degradation machinery was functional. Moreover, following *TRIBAL* suppression, *TRIB1* exhibited a significant correlation with *TRIBAL*

expression, suggesting that *TRIBAL* abundance was accurately measured. Rather than having no impact, *TRIBAL* ASO treatments were associated with highly variable TOI and *TRIBAL* abundance. This apparent noise may reflect genuine biological variation elicited by *TRIBAL* targeting.

Although HepG2 and HuH-7 cells were inadequate models for studying *TRIBAL*, the locus affected cell proliferation in these cells, as revealed by the selection of unedited or minimally edited populations. Since *TRIBAL* suppression did not reduce HepG2 proliferation (HuH-7s were not tested), we speculate that the locus may play additional roles independent of its transcript. For instance, the *TRIBAL* locus may facilitate TRIB1 expression, which promotes cell growth in both cell models, through enhancer-like effects [33,34]. The transformation of HepG2 and HuH-7 may also activate pathways that neutralize or antagonize *TRIBAL* metabolic functions.

Interestingly, native and recombinant *TRIBAL* transcripts differed profoundly in their sensitivity to RNAse R. Transduced *TRIBAL1* was uniquely sensitive to RNAse R, which may underlie its inability to protect against *TRIBAL* suppression or single-handedly regulate TOI expression in HepaRG (this study) or primary hepatocytes [1]. The proximal reasons for this difference are unknown. They are probably unrelated to transcript sequence, as *TRIBAL1* corresponds to the predominant form in HepG2 cells and is the only form identified in primary hepatocytes and HepaRG cells. Instead, we propose that the native locus is critical for accurate *TRIBAL* folding and imparting function. *TRIBAL* emerging from its native locus may mature differently from the transduced transcript through the participation of locus-enriched chaperones. Differential folding could be tested *in situ* using dedicated methodologies (rather than RNAse R sensitivity of isolated RNA) such as dimethyl sulfate RNA methylation or SHAPE (Selective 2′-Hydroxyl Acylation analyzed by Primer Extension) and its derivatives [35,36]. Moreover, locus-enriched alkylating enzymes may decorate the emerging transcript with chemical modifications that, in addition to affecting folding proper, may confer unique binding sites for cognate RNA binding proteins. Our understanding of RNA methylation is still in its infancy; however, evidence suggests that it plays a crucial role in defining the structure and function of lncRNA [37,38]. Future work will examine these possibilities.

## Supporting information

**S1 File. Uncropped blots**
(PDF)

**S2 File. Supplementary Materials.**
(DOCX)

**S1 Fig. *TRIBAL* activation by dCRISPRa in HuH-7 and HepG2 does not impact the transcript panel significantly.**
Cells were transfected for 72 h with a plasmid (dCAS9-VP160) encoding the dCAS9-CRISPRa construct and sgRNA sequences (sg4 or sg5) targeting the promoter region of *TRIBAL*. Each point represents a biological replicate. Statistical significance was assessed using a one-sample t-test using a theoretical control value of 1 in Prism. Nominal P values approaching (p ≤ to 0.1) or surpassing nominal significance are shown. *CYP7A1* abundance in HuH-7 cells was too low to be reproducibly measured.
(PDF)

**S2 Fig. Loss of *TRIBAL* is deleterious and selected against in HepG2 and HuH-7.** A, *TRIBAL* was targeted with CRISPR and 2 single guide RNAs flanking exon 1 (sg9 and sg14). The schema of sgRNA positions is shown at the top. The red arrow points to the position of the deleted allele. The experiment was repeated thrice (HepG2) or twice (HuH-7), with similar results. DNA from whole cell lysates was amplified with primers flanking the *TRIBAL* sgRNA cognate sites or targeting JunD (positive control). B, Impact of *TRIBAL* ASOs on HepG2 cell proliferation. A one-tailed t-test, performed under the hypothesis that *TRIBAL* suppression should reduce cell proliferation, showed no significant difference.
(PDF)

**S3 Fig.** ***TRIBAL*** **expression in hepatocyte cell models.** *TRIBAL* expression was measured in cell models treated with the CTLASO for 72 h. Each point represents a distinct biological replicate. Expression is expressed relative to *PPIA*. To simplify visualization, error bars below the mean are not shown.
(PDF)

**S4 Fig.** **Volcano plots of hepatocytes and HepaRG cells treated with** ***TRIBAL*** **ASO2.** Volcano plots of array data from hepatocytes and HepaRG cells treated with *TRIBAL* ASO2 (vs CTLASO). Nominal hits experiencing at least 2-fold absolute change in expression are colored.
(PDF)

**S5 Fig.** **Functional consistency between HepaRG cells and hepatocytes treated with** ***TRIBAL*** **ASO2.** A, Venn diagram of nominally impacted Transcript IDs (mapped to Entrez Genes) in HepaRG cells and hepatocytes. B, Venn diagram of FDR significant Gene Ontology terms identified by GSEA in *TRIBAL*-suppressed HepaRG cells and hepatocytes. HepaRG (magenta) and hepatocytes (blue). C, the ontologies from B were clustered with Revigo and exported into Cytoscape for visualization. Singletons were removed to aid visualization. A complete list is shown in S1-S2 Tables.
(PDF)

**S6 Fig.** **Functional overlap between** ***TRIBAL*** **and TRIB1 in HepaRG cells.** A, Venn diagram of nominally impacted Transcript IDs (mapped to Entrez Genes) in HepaRG cells targeted with *TRIBAL* or *TRIB1* ASO2. B, Venn diagram of FDR-significant Gene Ontology terms identified by GSEA in HepaRG cells targeted with *TRIBAL* or *TRIB1* ASO2. *TRIBAL* (magenta) and *TRIB1* (blue). C, categories from B were clustered with Revigo and exported into Cytoscape for visualization. Singletons were removed to aid visualization. See Tables S2-S3 for the complete list of ontologies.
(PDF)

**S7 Fig.** **Identification of** ***TRIBAL1*** **in HepaRG cells.** *TRIBAL* was amplified from cDNA derived from naïve, differentiated, HepaRG cells using PCR primers complementary to exon 1 and either exon 7 or Exon 2 (positive control) of *TRIBAL*. A BamHI digest was used to validate the PCR product as *TRIBAL1*, consisting of Exon 1, 2, and 7 of *TRIBAL*. Top, schematic of the PCR product and location of the BamHI site. Bottom, Agarose gels (1.5%) of the PCR products (left) and BamHI digest of the ~400 bp PCR product (right). * indicates primer dimers. The experiment was performed on 2 biological replicates with identical results.
(PDF)

**S8 Fig.** **Impact of** ***TRIBAL1*** **Transduction on HepaRG Transcripts of Interest.** HepaRG cells were transduced with PLVX*TRIBAL1* or PLVX for 72 h. RNA was then isolated and quantified by qRT-PCR. *TRIBAL* was upregulated by 6700 ± 4600 (S.D.). Values were internally normalized to PPIA and are expressed relative to the values from the PLVX transduced controls. Differences were not statistically different from the PLVX values (one-sample t-test using a theoretical control value of 1), except for *TRIBAL* ($p = 0.032$).
(PDF)

**S9 Fig.** ***TRIBAL1*** **transduction in HepaRG cells cannot protect from the impacts of** ***TRIBAL*** **suppression.** HepaRG cells were transduced with PLVX*TRIBAL1* or PLVX for 48 h before treatment with *TRIBAL* ASO2 or CTLASO for 72 h. RNA was then isolated, converted to cDNA, and analyzed for the indicated targets by qRT-PCR. *TRIBAL* transduction resulted in a 460-fold (± 350, S.D.) increase in *TRIBAL* abundance. Each point is a biological replicate.
(PDF)

**S10 Fig.** **Relative expression of the TOIs in the hepatocyte models.** Transcript levels assessed in CTLASO-treated HepG2, HuH-7, HepaRG, and primary hepatocytes. Statistical significance was tested using one-way ANOVA, followed by a post-hoc Tukey's multiple comparisons test. *, $p < 0.05$; **, $p < 0.01$; ***, $p < 0.001$, ****, $p < 0.0001$.
(PDF)

**S1 Table. GSEA of *TRIBAL*-suppressed hepatocytes. .** GSEA of hepatocytes treated with *TRIBAL* ASO2 vs CTLASO. 27947 Entrez gene IDs were submitted to the analysis, of which 14221 IDs were annotated to Gene Ontology (Biological noRedundant) categories and were used for the enrichment analysis. The top 200 hits are shown, arranged incrementally by normalized enrichment. Terms in bold are shared with HepaRG cells. FDR, false-discovery rate. Size: number of Entrez IDs populating the geneSet. LeadingEdge_Num: Number of leading-edge IDs (number of IDs that contributed the most to the enrichment). LeadingEdge_Entrez ID. NCBI Entrez gene IDs populating the LeadingEdge. (XLSX)

**S2 Table. GSEA of *TRIBAL*-suppressed HepaRG.** GSEA of HepaRG treated with *TRIBAL* ASO2 vs CTLASO. 27947 Entrez gene IDs were submitted to the analysis, of which 14221 IDs were annotated to Gene Ontology (Biological noRedundant) categories and were used for the enrichment analysis. The top 200 hits are shown, arranged by increasing normalized effect size. Terms in bold are shared with primary hepatocytes. FDR, false-discovery rate. Size: number of Entrez IDs populating the geneSet. LeadingEdge_Num: Number of leading-edge IDs (number of IDs that contributed the most to the enrichment). LeadingEdge_Entrez ID. NCBI Entrez gene IDs populating the LeadingEdge. (XLSX)

**S3 Table. Shared and unique *TRIBAL* FDR terms in HepaRG cell and hepatocytes.** (XLSX)

**S4 Table. GSEA of *TRIB1*-suppressed HepaRG.** GSEA of HepaRG treated with *TRIB1* ASO vs CTLASO. 27947 Entrez gene IDs were submitted to the analysis, of which 14221 IDs were annotated to Gene Ontology (Biological noRedundant) categories and were used for the enrichment analysis. The top 200 hits are shown, arranged by increasing normalized effect size. FDR, false-discovery rate. Size: number of Entrez IDs populating the geneSet. LeadingEdge_Num: Number of leading-edge IDs (number of IDs who contributed the most to the enrichment). LeadingEdge_Entrez ID. NCBI Entrez gene IDs populating the LeadingEdge. (XLSX)

**S5 Table. Shared and unique *TRIB1* and *TRIBAL* FDR terms in HepaRG cells.** (XLSX)

**S6 Table. GSEA of *TRIBAL* vs *TRIB1* in HepaRG.** 27947 Entrez gene IDs were submitted to the analysis, of which 14221 IDs were annotated to Gene Ontology (Biological noRedundant) categories and were used for the enrichment analysis. Top 200 hits are shown, arranged by increasing normalized effect size. FDR-significant hits are highlighted. FDR, false-discovery rate. Size: number of Entrez IDs populating the geneSet. LeadingEdge_Num: Number of leading-edge IDs (number of IDs who contributed the most to the enrichment). LeadingEdge_Entrez ID. NCBI Entrez gene IDs populating the LeadingEdge. (XLSX)

## Author contributions

**Data curation:** Sébastien Soubeyrand.

**Formal analysis:** Sébastien Soubeyrand.

**Funding acquisition:** Sébastien Soubeyrand, Ruth McPherson.

**Investigation:** Sébastien Soubeyrand, Paulina Lau.

**Methodology:** Sébastien Soubeyrand.

**Project administration:** Sébastien Soubeyrand, Ruth McPherson.

**Resources:** Paulina Lau.

**Supervision:** Sébastien Soubeyrand.

**Validation:** Sébastien Soubeyrand.

**Visualization:** Sébastien Soubeyrand.

**Writing – original draft:** Sébastien Soubeyrand.

**Writing – review & editing:** Sébastien Soubeyrand, Ruth McPherson.

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
