## [Decision Letter · Decision Letter 0]

30 Apr 2025

Dear Dr. Soubeyrand,

Thank you for submitting your manuscript to PLOS ONE. After careful consideration, we feel that it has merit but does not fully meet PLOS ONE’s publication criteria as it currently stands. Therefore, we invite you to submit a revised version of the manuscript that addresses the points raised during the review process.

We look forward to receiving your revised manuscript.

Kind regards,

Zheng Yuan

Academic Editor

PLOS ONE

“This work was supported by the Canadian Institutes of Health Research (CIHR) FDN-154308 (RM).”

“This work was supported by the Canadian Institutes of Health Research (CIHR) FDN-154308 (RM).

https://cihr-irsc.gc.ca/e/193.html

The funder did not play play any role in the study design, data collection and analysis, decision to publish, or preparation of the manuscript”

3. Please note that your Data Availability Statement is currently missing the repository name and/or the DOI/accession number of each dataset OR a direct link to access each database. If your manuscript is accepted for publication, you will be asked to provide these details on a very short timeline. We therefore suggest that you provide this information now, though we will not hold up the peer review process if you are unable.

Reviewers' comments:

Reviewer's Responses to Questions

**Comments to the Author**

1. Is the manuscript technically sound, and do the data support the conclusions?

Reviewer #1: Partly

Reviewer #2: Yes

Reviewer #3: Partly

2. Has the statistical analysis been performed appropriately and rigorously?

Reviewer #1: N/A

Reviewer #2: Yes

Reviewer #3: Yes

3. Have the authors made all data underlying the findings in their manuscript fully available?

Reviewer #1: No

Reviewer #2: Yes

Reviewer #3: No

4. Is the manuscript presented in an intelligible fashion and written in standard English?

Reviewer #1: No

Reviewer #2: Yes

Reviewer #3: Yes

Reviewer #1: In this study, Sébastien and colleagues used an unconventional system to investigate the role of the lncRNA TRIBAL in hepatocyte function. However, I was unable to locate the main figures throughout the entire manuscript provided, which makes it difficult to conduct a thorough review. I would appreciate clarification to ensure there were no submission errors. Below are a few general comments that may help improve the manuscript upon resubmission or revision:

1. Terms such as “ASO” should be written in full upon first mention. Several other abbreviations also appear without prior definition and should be clarified for reader comprehension.

The manuscript does not introduce or describe the differences between ASO1 and ASO2. It is unclear how the authors interpret distinct phenotypes arising from the use of these two distinct ASOs.

2. The overall readability of the manuscript is low. I strongly recommend that the authors seek input from a professional editor or experienced colleague to improve the clarity and flow of the writing. Several expressions are vague, making it difficult to follow the intended conclusions.

3. The figures and their legends require substantial improvement. For instance, terms such as “VP160/CG4” and “VP160/CG5” appear in the figures without any explanation in the legends and the content. I suggest the authors have project-unrelated colleagues review the figures and manuscript to ensure all components and conclusions are clearly understandable prior to resubmission.

Reviewer #2: This article the function of lncRNA TRIBAL in the hepatocyte models, with focus on its regulation lo liver-specific transcriptional programs and then compare its function to TRIB1. Overall the experiments are designed well and excuted in a proper way. I recommend acceptance after minor revision to improve scientific clarity and completeness.

1. In the main text, the authors state that "TRIBAL1 did not protect against ASO2" and that "these experiments suggested that overexpressed TRIBAL1 was dysfunctional." (Row 330). Since this observation is an important part of the study, it would be better to briefly confirm whether the TRIBAL transcript endogenously expressed in HepaRG cells and primary hepatocytes matches the TRIBAL1 sequence used for overexpression. This could be achieved by qPCR spanning exon junctions, or by referencing existing public database annotations.

2.The authors propose that differences in RNA structure might explain the inability of recombinant TRIBAL1 to mimic native TRIBAL function, based on RNAse R resistance assays (Strikingly, recombinant TRIBAL1 was highly sensitive to RNAseR, on par with the PPIA mRNA. By contrast, native TRIBAL, like U1, a highly structured snRNA, showed a near-complete resistance to RNAseR, Row 336-338). While this is a reasonable and interesting hypothesis, it remains speculative without direct structure mapping. It would strengthen the scientific rigor of the Discussion if the authors explicitly acknowledge this and suggest that future studies using RNA structure probing methods (such as SHAPE Selection to probe 2nd structure) could further confirm structural differences.

While the manuscript addresses a relatively focused question, it does so with careful methodology and clear data presentation. The study will be a useful resource for researchers investigating lncRNA function and hepatocyte models.

Reviewer #3: Soubeyrand et al. present a follow-up to their previous work on long non-coding RNA TRIBAL/TRIB1AL. The authors successfully identify HepaRG cells as a suitable model system for mechanistic studies, addressing the limitations of previously used cell lines. They also elucidate the mechanism by which TRIBAL regulates key hepatocyte genes involved in metabolic function. I would like to offer the following constructive feedback:

Major points for consideration:

1. The manuscript format needs adjustment - the main figures do not appear to be integrated with the text (only supplementary figures are visible). This formatting issue makes it challenging to connect specific statements to their corresponding data, affecting the manuscript's clarity and readability.

2. The study would benefit from additional experiments to further strengthen its novelty and impact.

Minor points for consideration:

1. The introduction would be strengthened by several improvements. For instance, citing more recent literature on lncRNA functionality would provide a more current context than the 10-year-old Guttman et al. study. Quantitative statements with specific statistics would be preferable to qualitative terms like "likely." Additionally, reorganizing the introduction to provide general background on lncRNAs before focusing on TRIBAL would improve the logical flow.

2. Some statistical analyses may need additional clarification. For example, in Figure S1, certain groups appear to have only 3 samples, which may impact statistical robustness.

3. The conclusion stating that "TRIBAL suppression in HepaRG resulted in ~5 times fewer nominally significant hits than in hepatocytes, suggesting either the possible loss of TRIBAL targets in HepaRG or lower quality data" could be strengthened. If data quality is a concern, the rationale for using this dataset/model should be further justified.

4. For the GO analysis, could you please provide additional details regarding p-values, the number of genes in each set, and any analysis of gene overlap between categories?

The observation that TRIBAL transduction did not restore function in TRIBAL-suppressed cells warrants further discussion. 5. Could the authors elaborate on whether this affects TRIBAL's potential as a therapeutic target? Additionally, addressing the structural differences and their potential causes would be valuable.

**Do you want your identity to be public for this peer review?** For information about this choice, including consent withdrawal, please see our Privacy Policy

Reviewer #1: No

Reviewer #2: No

Reviewer #3: No

---

## [Author Response · Author response to Decision Letter 1]

23 May 2025

Reviewer #1: In this study, Sébastien and colleagues used an unconventional system to investigate the role of the lncRNA TRIBAL in hepatocyte function. However, I was unable to locate the main figures throughout the entire manuscript provided, which makes it difficult to conduct a thorough review. I would appreciate clarification to ensure there were no submission errors. Below are a few general comments that may help improve the manuscript upon resubmission or revision:

We apologize for this oversight. Figures were indeed inexplicably omitted during submission (rather the suppl. Figures seemed to have been submitted twice). This has been corrected. As a result, clarity and readability will no doubt be improved. Again, our apologies.

1. Terms such as “ASO” should be written in full upon first mention. Several other abbreviations also appear without prior definition and should be clarified for reader comprehension.

Abbreviations should be defined on first appearance.

The manuscript does not introduce or describe the differences between ASO1 and ASO2. It is unclear how the authors interpret distinct phenotypes arising from the use of these two distinct ASOs.

The ASOs were introduced in our previous publication on TRIBAL, wherein we described their overall convergence but some apparent differences. A reference has been added (line 238) and the differences between the ASOs described in the Methods section (Under “Antisense Oligonucleotides”) and in the Discussion section.

2. The overall readability of the manuscript is low. I strongly recommend that the authors seek input from a professional editor or experienced colleague to improve the clarity and flow of the writing. Several expressions are vague, making it difficult to follow the intended conclusions.

Possibly vague expressions were identified and corrected. With the essential inclusion of figures, clarity and flow should be markedly improved.

3. The figures and their legends require substantial improvement. For instance, terms such as “VP160/CG4” and “VP160/CG5” appear in the figures without any explanation in the legends and the content. I suggest the authors have project-unrelated colleagues review the figures and manuscript to ensure all components and conclusions are clearly understandable prior to resubmission.

Thank you for your suggestion. Figure legends are now expanded to include more information.

Reviewer #2: This article the function of lncRNA TRIBAL in the hepatocyte models, with focus on its regulation lo liver-specific transcriptional programs and then compare its function to TRIB1. Overall the experiments are designed well and excuted in a proper way. I recommend acceptance after minor revision to improve scientific clarity and completeness.

1. In the main text, the authors state that "TRIBAL1 did not protect against ASO2" and that "these experiments suggested that overexpressed TRIBAL1 was dysfunctional." (Row 330). Since this observation is an important part of the study, it would be better to briefly confirm whether the TRIBAL transcript endogenously expressed in HepaRG cells and primary hepatocytes matches the TRIBAL1 sequence used for overexpression. This could be achieved by qPCR spanning exon junctions, or by referencing existing public database annotations.

The TRIBAL transcript used corresponded to the only form (TRIBAL1) that we previously identified in primary hepatocytes by RACE experiments. This point is now made explicit in the Results section (line 408) as well. As suggested, we have performed PCR validation in HepaRG cells using oligonucleotides targeting the first and last exon of TRIBAL1 form and demonstrate that a single band was obtained, corresponding to the predicted size of TRIBAL1. Its identity was further confirmed by restriction digest. These results indicate that TRIBAL1 is indeed the primary form in HepaRG cells. These new results have been added as S7 Fig and mentioned in the Discussion (line 519)

2.The authors propose that differences in RNA structure might explain the inability of recombinant TRIBAL1 to mimic native TRIBAL function, based on RNAse R resistance assays (Strikingly, recombinant TRIBAL1 was highly sensitive to RNAseR, on par with the PPIA mRNA. By contrast, native TRIBAL, like U1, a highly structured snRNA, showed a near-complete resistance to RNAseR, Row 336-338). While this is a reasonable and interesting hypothesis, it remains speculative without direct structure mapping. It would strengthen the scientific rigor of the Discussion if the authors explicitly acknowledge this and suggest that future studies using RNA structure probing methods (such as SHAPE Selection to probe 2nd structure) could further confirm structural differences.

We agree with the reviewer. This point is further raised in the Discussion, as suggested.

While the manuscript addresses a relatively focused question, it does so with careful methodology and clear data presentation. The study will be a useful resource for researchers investigating lncRNA function and hepatocyte models.

Reviewer #3: Soubeyrand et al. present a follow-up to their previous work on long non-coding RNA TRIBAL/TRIB1AL. The authors successfully identify HepaRG cells as a suitable model system for mechanistic studies, addressing the limitations of previously used cell lines. They also elucidate the mechanism by which TRIBAL regulates key hepatocyte genes involved in metabolic function. I would like to offer the following constructive feedback:

Major points for consideration:

1. The manuscript format needs adjustment - the main figures do not appear to be integrated with the text (only supplementary figures are visible). This formatting issue makes it challenging to connect specific statements to their corresponding data, affecting the manuscript's clarity and readability.

As mentioned above (Reviewer 1), figures were inadvertently omitted. We apologize for this oversight. Inclusion of the figures should significantly improve the clarity and readability of the manuscript.

2. The study would benefit from additional experiments to further strengthen its novelty and impact.

We have included additional data in response to the reviewer (Volcano plot and TRIBAL characterization in HepaRG cells). As for additional replication of presented data, see point 2 below.

Minor points for consideration:

1. The introduction would be strengthened by several improvements. For instance, citing more recent literature on lncRNA functionality would provide a more current context than the 10-year-old Guttman et al. study. Quantitative statements with specific statistics would be preferable to qualitative terms like "likely." Additionally, reorganizing the introduction to provide general background on lncRNAs before focusing on TRIBAL would improve the logical flow.

Thank you for this suggestion. The introduction has been expanded and now includes additional and more up-to-date references.

2. Some statistical analyses may need additional clarification. For example, in Figure S1, certain groups appear to have only 3 samples, which may impact statistical robustness.

Most of our experiments were performed 3 or 4 times by design, as we were expecting to observe large effect sizes, as we previously observed with primary hepatocytes. We have not attempted to add additional rounds of experiments, as these experiments were consistent with a null hypothesis. However, we readily acknowledge that with a nominal number of repeats, power is low and that subtle but genuine differences (albeit of uncertain biological significance) could be missed.

3. The conclusion stating that "TRIBAL suppression in HepaRG resulted in ~5 times fewer nominally significant hits than in hepatocytes, suggesting either the possible loss of TRIBAL targets in HepaRG or lower quality data" could be strengthened. If data quality is a concern, the rationale for using this dataset/model should be further justified.

After re-examining our data (more specifically the volcano plots), it became apparent that the main difference between HepaRG and Hepatocytes was related to the fold-change, not the p-values (i.e. changes in HepaRG were of much lower magnitude). Thus, we are reasonably confident that lower quality data is not an issue but that HepaRGs were less impacted. This sentence has been modified and the volcano plots are now included (as S4. Fig; Fig S4 has been moved to S5 etc.)

4. For the GO analysis, could you please provide additional details regarding p-values, the number of genes in each set, and any analysis of gene overlap between categories?

The observation that TRIBAL transduction did not restore function in TRIBAL-suppressed cells warrants further Discussion.

The p-values are available in the Supplemental Tables. P=values were obtained, as is typical for GSEA, by permutation-based analysis of ranked lists. The p-value was calculated by determining how frequently the ES from the actual ranking is greater than that for the random permutations. WebGestalt uses a typical 1000 permutations to calculate the significance of the enrichment score (ES). This has been included in the Methods section. The tables were expanded to include the number of genes in each set (rather than just the leading edge transcripts). Although the gene overlap between categories is not shown (given the large number of categories), the ENTREZ gene IDs populating these categories are now included in the tables.

5. Could the authors elaborate on whether this affects TRIBAL's potential as a therapeutic target? Additionally, addressing the structural differences and their potential causes would be valuable.

We do not believe that TRIBAL targeting is a viable therapeutic approach, on both theoretical and practical grounds. For one, as TRIBAL is important to sustain normal liver function, TRIBAL suppression is predicted to be toxic. As for its overexpression, for example as a treatment for steatotic liver disease, our data indicate that this is not feasible since transduced TRIBAL1 cannot impact the targets of TRIBAL. Assuming that transcriptional activation approaches may be optimized to activate the endogenous locus, we note that genetic predisposition to higher TRIBAL abundance associate with increased cardiovascular risk, suggesting that higher TRIBAL expression may be detrimental. On practical grounds, a major limitation resides in the paucity of actionable animal models, as TRIBAL is only expressed in primates. Although one could envision hybrid cancer models (e.g. human tumors in mouse models), the lack of in vitro impact of the ASOs on HuH-7, HepG2, and undifferentiated HepaRGs suggests that (liver) cancers may not be responsive to TRIBAL targeting.

We have expanded our Discussion (last paragraph) to suggest dedicated approaches to test structural differences between the transduced and endogenous forms. Although we propose models to account for these differences, they remain speculative until structural differences are confirmed.

---

## [Decision Letter · Decision Letter 1]

17 Jun 2025

Dear Dr. Soubeyrand,

Thank you for submitting your manuscript to PLOS ONE. After careful consideration, we feel that it has merit but does not fully meet PLOS ONE’s publication criteria as it currently stands. Therefore, we invite you to submit a revised version of the manuscript that addresses the points raised during the review process.

We look forward to receiving your revised manuscript.

Kind regards,

Zheng Yuan

Academic Editor

PLOS ONE

Reviewers' comments:

Reviewer's Responses to Questions

**Comments to the Author**

Reviewer #1: (No Response)

2. Is the manuscript technically sound, and do the data support the conclusions?

Reviewer #1: Partly

3. Has the statistical analysis been performed appropriately and rigorously?

Reviewer #1: Yes

4. Have the authors made all data underlying the findings in their manuscript fully available?

Reviewer #1: Yes

5. Is the manuscript presented in an intelligible fashion and written in standard English?

Reviewer #1: No

Reviewer #1: This manuscript by Soubeyrand et al. aims to elucidate the role of the TRIBAL in hepatocyte models, focusing on its regulatory relationship with TRIB1 and its impact on liver-specific transcriptional programs. While the study presents a substantial amount of data, however, I find several issues that limit the current conclusions and overall impact of this work.

Major Points:

1. If the TOIs were selected based on previous publications, please cite the relevant references. Additionally, the title for the first paragraph of the result seems inappropriate—I did not find any content specifically addressing the optimal growth of HepG2 or HuH-7 cells. Although the authors mention, in the Methods section that they have explained the differences between ASO1 and ASO2 (and additional ASOs in figure 4), it remains unclear what drives the phenotypic differences. Are ASO1 and ASO2 targeting different functional domains, leading to distinct biological effects? Or is the difference purely due to targeting efficiency? Please summarize this in a concise sentence to help readers understand the nature of these ASOs.

2. What are the baseline expression levels of the TOIs in each cell line used in the study (HepG2, Huh-7 vs HepaRG)? Are there inherent expression differences without TRIBAL manipulation?

3. ASO9 appears to exhibit stronger suppression than ASO2, outperforming it in some assays. Therefore, ASO9 should be included as a repeat along with ASO2 in subsequent experiments to validate findings, particularly when testing TRIBAL1 functionality.

4. Comparing Figures 2, 4, and 5 reveals inconsistencies in the effects of ASO2 on TRIB1 expression. In Figures 2 and 5, ASO2 does not significantly affect TRIB1 expression in HepaRG cells, whereas in Figure 4, the effect appears significant. Moreover, published data using primary hepatocytes also show no significant reduction in TRIB1 with ASO2 treatment. These discrepancies call into question the conclusion presented in lines 288–291. Additionally, in Figure 5B, suppression of TRIB1 using either ASO1 or ASO2 does not alter TRIBAL expression. Do you believe the regulation of TOIs by TRIBAL suppression is independent or dependent on TRIB1? Please clarify.

5. In Figure 6, a correlation between TRIBAL and TRIB1 expression does not imply co-regulation. The conclusion that TRIBAL regulates TRIB1 based on expression correlation is speculative and not mechanistically justified.

Minor points:

1. Some figures can be combined to improve reading continuity—for example, Figures 2 and 3 could be merged.

2. Figure legends and tables are currently embedded within the main text; they should be formatted appropriately according to journal guidelines.

**Do you want your identity to be public for this peer review?** For information about this choice, including consent withdrawal, please see our Privacy Policy

Reviewer #1: No

---

## [Author Response · Author response to Decision Letter 2]

17 Jul 2025

Reviewer #1: This manuscript by Soubeyrand et al. aims to elucidate the role of the TRIBAL in hepatocyte models, focusing on its regulatory relationship with TRIB1 and its impact on liver-specific transcriptional programs. While the study presents a substantial amount of data, however, I find several issues that limit the current conclusions and overall impact of this work.

Thank you for a very detailed review. We have addressed specific concerns as detailed below and have rewritten several parts of the manuscript to improve clarity and readability.

Major Points:

1. If the TOIs were selected based on previous publications, please cite the relevant references. Additionally, the title for the first paragraph of the result seems inappropriate—I did not find any content specifically addressing the optimal growth of HepG2 or HuH-7 cells. Although the authors mention, in the Methods section that they have explained the differences between ASO1 and ASO2 (and additional ASOs in figure 4), it remains unclear what drives the phenotypic differences. Are ASO1 and ASO2 targeting different functional domains, leading to distinct biological effects? Or is the difference purely due to targeting efficiency? Please summarize this in a concise sentence to help readers understand the nature of these ASOs.

The relevant references have been added. Regarding optimal growth, we aimed to convey the meaning of optimal under normal growth conditions (standard media), rather than the optimal growth condition itself. For clarity, ‘optimal’ has been removed.

We have expanded the Methods section (Antisense oligonucleotides) to include a more complete description of the ASOs. As explained in that section, ASO1 targets exon 1 of TRIBAL, whereas all the others target either of 2 introns. We did find that ASO1 was modestly less effective at reducing TRIBAL levels, but significantly less effective at suppressing the TOIs. We previously demonstrated (as mentioned in the Discussion lines 496-512) that an additional ASO targeting exon 1 (not used in this work) was shown to have a comparable impact as ASO2 but resulted, on average, in a more modest ASO1-like impact in primary hepatocytes. Thus, we hypothesize that intron targeting has additional effects related to targeting of the nascent transcript, as noted in the Discussion.

2. What are the baseline expression levels of the TOIs in each cell line used in the study (HepG2, Huh-7 vs HepaRG)? Are there inherent expression differences without TRIBAL manipulation?

An important point. We have added a direct comparison amongst all the TOIs in HepG2, HuH-7, HepaRG and primary hepatocytes, obtained in CTLASO-treated cells. Note that these values were normalized to PPIA expression (which we have used as a robust housekeeper in our work over the years and is unaffected by TRIBAL suppression based on transcription array evidence). The additional figure (S10 Fig) was integrated intp the Discussion (line 485), where it seemed to fit best.

3. ASO9 appears to exhibit stronger suppression than ASO2, outperforming it in some assays. Therefore, ASO9 should be included as a repeat along with ASO2 in subsequent experiments to validate findings, particularly when testing TRIBAL1 functionality.

ASO9 may have a greater impact than ASO2 (Figure 4) on some transcripts (CYP7A1, HGMGCS2, ACAT1), but the difference is small (median 1.1-fold, average 1.2-fold). Thus, given the already sizeable impact of ASO2 and the lack of rescue with TRIBAL1 OE, we believe that repeated experiments with ASO9 are unlikely to inform on the main point, i.e. the incapacity of TRIBAL1 to rescue the impact of ASO-mediated TRIBAL knockdown.

4. Comparing Figures 2, 4, and 5 reveals inconsistencies in the effects of ASO2 on TRIB1 expression. In Figures 2 and 5, ASO2 does not significantly affect TRIB1 expression in HepaRG cells, whereas in Figure 4, the effect appears significant. Moreover, published data using primary hepatocytes also show no significant reduction in TRIB1 with ASO2 treatment. These discrepancies call into question the conclusion presented in lines 288–291. Additionally, in Figure 5B, suppression of TRIB1 using either ASO1 or ASO2 does not alter TRIBAL expression. Do you believe the regulation of TOIs by TRIBAL suppression is independent or dependent on TRIB1? Please clarify.

Our data demonstrates that TRIB1 expression was overall reduced, albeit not always enough to achieve statistical significance in any single round of 3-4 experiments. In Figure 2, an approximately 25% reduction, but not significant (1 out of 4 may be an outlier). In Figure 5 (hepatocytes), the reduction is also approximately 20%, and in Figure 4, the impact is more pronounced, on average, at 40%. Thus, on aggregate, TRIBAL suppression correlates with reduced TRIB1 expression.

However, we believe that the regulation of the TOIs by TRIB1 and TRIBAL occurs largely independently, based on ASO evidence showing that (as rightly pointed out), TRIBAL suppression can impact TOI, even when TRIB1 expression is modestly (or not) impacted (Fig 5A, TRIBAL ASO2). Similarly, TRIB1 suppression impacts the TOI even with minimal changes to TRIBAL. However, this apparent independence of function should not be misconstrued to imply that they act on different targets. This point has been clarified in the first two paragraphs of the Discussion.

5. In Figure 6, a correlation between TRIBAL and TRIB1 expression does not imply co-regulation. The conclusion that TRIBAL regulates TRIB1 based on expression correlation is speculative and not mechanistically justified.

We agree that a conclusion to the effect that TRIBAL can regulate TRIB1 was probably too speculative and have revised the text accordingly. The correlation is consistent (but not necessarily causative) with both transcripts sharing similar regulators (what we intended to convey with the phrase “co-regulation”). The Results section has been rewritten accordingly (lines 337-339).

Minor points:

1. Some figures can be combined to improve reading continuity—for example, Figures 2 and 3 could be merged.

Thank you for your suggestion. However, we feel that keeping RNA and protein results separately reduces confusion. The flow should improve significantly once the manuscript reaches the galley proof stage and the text-embedded legends are removed.

2. Figure legends and tables are currently embedded within the main text; they should be formatted appropriately according to journal guidelines.

Our understanding is that, according to the PLOSONE journal guidelines, tables and legends should be integrated within the submitted manuscript’s main text.

---

## [Decision Letter · Decision Letter 2]

11 Aug 2025

Deciphering the role of the lncRNA TRIBAL in hepatocyte models

PONE-D-25-17281R2

Dear Dr. Soubeyrand,

We’re pleased to inform you that your manuscript has been judged scientifically suitable for publication and will be formally accepted for publication once it meets all outstanding technical requirements.

Kind regards,

Zheng Yuan

Academic Editor

PLOS ONE

Additional Editor Comments (optional):

Reviewers' comments:

Reviewer's Responses to Questions

**Comments to the Author**

Reviewer #1: All comments have been addressed

2. Is the manuscript technically sound, and do the data support the conclusions?

Reviewer #1: Yes

3. Has the statistical analysis been performed appropriately and rigorously?

Reviewer #1: Yes

4. Have the authors made all data underlying the findings in their manuscript fully available?

Reviewer #1: Yes

5. Is the manuscript presented in an intelligible fashion and written in standard English?

Reviewer #1: Yes

Reviewer #1: The revised version addresses my previous concerns and overall meets the standard for acceptance. I endorse this work for publication.

**Do you want your identity to be public for this peer review?** For information about this choice, including consent withdrawal, please see our Privacy Policy

Reviewer #1: No

---

## [Editor Report · Acceptance letter]

PONE-D-25-17281R2

PLOS ONE

Dear Dr. Soubeyrand,

I'm pleased to inform you that your manuscript has been deemed suitable for publication in PLOS ONE. Congratulations! Your manuscript is now being handed over to our production team.

Kind regards,

on behalf of

Dr. Zheng Yuan

Academic Editor

PLOS ONE